# Evaluating Nutrient Intake of Career Firefighters Compared to Military Dietary Reference Intakes

**DOI:** 10.3390/nu12061876

**Published:** 2020-06-23

**Authors:** Brittany V.B. Johnson, John M. Mayer

**Affiliations:** U.S. Spine & Sport Foundation, San Diego, CA 92111, USA; johnmayer@usssf.com

**Keywords:** first responders, nutrition, tactical athletes

## Abstract

The primary goals of the Dietary Reference Intakes (DRI) are to plan and assess nutrient intakes to promote health, reduce chronic disease, and prevent toxicity. Firefighters have unique nutrient needs compared to the public due to their job demands. The military provides the only published guidance for tactical athletes’ nutrient needs. The purpose of this study was to determine whether firefighters were meeting the Military Dietary Reference Intakes (MDRI). A cross-sectional study was conducted in a sample of career firefighters (*n* = 150, 37.4 ± 8.4 year-old males) employed in Southern California. Data were gathered during baseline assessments from a Federal Emergency Management Agency-funded Firefighter Wellness Initiative. Participants were asked to log their food and beverage consumption over a 72-h period. Descriptive statistics (means, standard deviations, 95% confidence intervals) were calculated for all participant characteristics and average three-day nutrient intakes. A 95% confidence interval compared their nutrient intake to MDRI to identify differences in nutrient intakes, significance accepted at *p* = 0.05. Compared to MDRI reference values, firefighters consumed an inadequate amount of total calories, linolenic and alpha-linolenic fatty acid, fiber, vitamins D, E, and K, potassium, magnesium, zinc, and carbohydrates. Vitamin D, magnesium, and potassium had the greatest shortcomings (95.3%, 94.0%, and 98.7%, respectively, under MRDA). Thus, firefighters are not meeting the established MDRI for several key nutrients required to promote health, improve performance, and reduce chronic disease. Dietitians and health care providers may use the results of this study to help design health promotion programs for this population. Future research should develop a customized reference intake for firefighters.

## 1. Introduction

Dietary Reference Intakes (DRI) were developed by expert panels and committees from the Food and Nutrition Board of the Institute of Medicine. The primary goals of the DRI are to plan and assess nutrient intakes for specific age and gender needs to promote health, reduce chronic disease, and prevent toxicity [1]. The DRI influence national nutrition policies and can be a tool for assessing individual nutrient adequacy. These standards are intended to guide general populations; however, their use for specialized populations, such as tactical athletes, are limited. Tactical athletes are individuals in law enforcement, military, and first responders (e.g., firefighters) who require physical training for occupational performance [2]. Firefighting is one of the most hazardous, physically demanding, and psychologically stressful occupations, requiring physical exertion while wearing heavy protective gear [3]. To our knowledge, the military provides the only published guidance for reference intakes (MDRI) specifically for tactical athletes’ demands [4,5]. However, the suitability of MDRI for use outside of the military is unknown. Firefighters are tactical athletes who could benefit from customized DRI to meet specific job demands, improve performance, enhance resiliency and recovery, and promote optimal health. No standards are available to guide clinical decision-making in prescribing dietary and nutrition plans for firefighters. Comparing firefighters’ nutrient intake to the DRI of the general population is not appropriate, given their unique nutritional demands to maintain physical performance, recreational fitness, injury prevention, and improve their quality of life. Thus, research is needed to identify nutritional habits in firefighters’ diets so that nutrition interventions can be developed to improve their quality of life and job performance.

To date, most studies have analyzed firefighters’ nutritional intake using the Food Frequency Questionnaire (FFQ). Very few have used three-day food records, and they also have small sample sizes [6,7,8]. Three-day food records are deemed valid for average intakes when compared to longer durations, and reduce survey fatigue [9]. Although much research has focused on intervention-based studies with firefighters, little is known about the adequacy of nutrient intakes compared to standard DRI or MDRI. The MDRI were developed for planning and developing feeding menus during bootcamp and rations for military operations to meet the established nutrition guidelines. The MDRI nutrient requirements were adjusted from the DRI for personnel undergoing various operational conditions [1,10]. The MDRI were established for 17 to 50-year-old military men who are physically active [11]. While a small minority of active-duty firefighters are over 50, the majority fall into this age category designed for the military personnel. Regardless, firefighters’ job demands and requirements are no different among age groups.

The MDRI were developed with the understanding that job-specific demands would require more nutrients for athletic performance compared to the general population. Recommendations for the DRI were developed based on healthy individuals undergoing light activities. Therefore, with increased physical activity, athletes and tactical athletes would need additional amounts of some nutrients to support performance demands [12]. The MDRI have increased reference intakes for nutrients supported by research that suggest requirements are impacted by physical activity and environmental factors (e.g., hot or cold temperatures) while undergoing military operations. The nutrients with a higher reference compared to DRI are folate, vitamin C, calcium, iron, magnesium, phosphorus, and selenium. Given this evidence, the MDRI are more appropriate for measuring firefighters’ nutrient intake to support health and performance. Further, MDRI are appropriate for comparison in our study because the MDRI are intended to be averaged over a three-day period [11]. To our knowledge, current research has not assessed or evaluated firefighters’ nutrient intake compared to MDRI based on their higher physical demands. The purpose of this study was thus to compare nutrient intakes of career firefighters to the MDRI.

## 2. Materials and Methods

The population of interest was career firefighters in the United States. A convenience sample of career firefighters who were enrolled in the Regional Firefighter Wellness Initiative was enrolled in this study. The study sample consisted of all participants who completed the three-day food record. Given the descriptive nature of the current study, a sample size of *n* = 150 was adequate to achieve the study’s aims. The selected sample size is larger than other studies assessing nutritional intake in firefighters and military personnel [10,11,13,14].

A cross-sectional study was conducted in a sample of career firefighters as part of baseline assessments within a Regional Firefighter Wellness Initiative [15]. The study was conducted with 13 career fire departments in Southern California, U.S., ranging from urban, coastal, native American reservation, and national border departments. Full-duty career firefighters (*n* = 150, males) were enrolled in this study. The shift schedule within the participating fire departments were either 4 s and 6 s (24 h shift) or 48–96 (48 h shift). The participants had a variety of eating habits prior to enrollment. Inclusion criteria to participate were: (1) career firefighters, full active duty, and no work restrictions. Exclusion criteria were: (1) relevant current workers’ compensation or personal injury case, (2) pregnant women, and (3) any reason the principal investigator deemed unsuitable, such as clinical contraindications to physical activity, behavioral concerns, and unwillingness to complete the study procedures. All firefighters who completed the three-day food record were included in this study. The research procedures were approved by an Institutional Review Board and all firefighters provided informed consent prior to voluntary participation (Aspire IRB—a WIRB-Copernicus Group company, San Diego, CA, USA; Protocol number: USSSF201901; Approval Date: 25 February 2019).

Recruitment occurred from June 2019 to August 2019 and consisted primarily of face-to-face presentations. During recruitment presentations, research personnel introduced the Regional Firefighter Wellness Initiative to candidates. Firefighters who expressed interest were given consent to participate following the presentations, or provided their personal information to be contacted later and scheduled an appointment. The individual firefighter was provided an opportunity to consent in a private area of the fire station if preferred. Research personnel thoroughly explained the purpose of the study, risks/benefits, confidentiality, and that their participation was voluntary. At any point, the firefighter could choose to withdraw from the study for any reason or no reason, without penalty. After consent, participants completed a battery of baseline assessments, which included self-report questionnaires, anthropometric measures, and physical fitness tests as part of the overlying grant. Self-report questionnaires for demographics, health history, injury and illness, physical function, disability, behavioral, physical activity, exercise habits, diet habits, stages of change, sleep questionnaires, and a food record were obtained.

There are several subjective assessments for dietary and nutritional intakes that include a 24 h recall, dietary record, dietary history, or FFQ. The three-day food record was selected as the primary method to obtain dietary intake due to its strengths, such as recording actual intakes rather than food group servings, as well as real-time data collection, which reduces recall bias [16]. Previous dietary assessments within the population focused on FFQs rather than detailed nutrient intakes with a food record. While three days is a relatively short duration to collect dietary intakes, when compared to longer durations, similar intakes were observed and deemed valid for averaging intakes [9]. Therefore, to capture a specific nutrient intake following baseline assessments, the firefighters completed a three-day food record via the online system: FoodProdigy (Version 1.81, ESHA Research, Salem, OR, USA).

FoodProdigy is an online food logging database where the firefighter submits their food record over three days via a personal account. To improve the accuracy of recording dietary intakes, firefighters were provided a protocol with detailed instructions on how to accurately measure food and beverages. Capturing dietary intake in this manner has been shown to be reliable and valid [17]. The protocol asked firefighters to measure their food using measuring cups and a food scale when possible. If measuring or weighing food was not possible, instructions and a handout were provided with standard portions using simple comparators (e.g., baseball = 1 cup broccoli) or their hands (e.g., palm size = 3 oz. protein foods) to estimate portions [18]. Due to the limitation of varying hand sizes, firefighters were encouraged to use the standardized approach using comparators such as a deck of cards, a baseball, or dice when estimating portions. Participants were asked to log their food and beverages over a 72 h period (two on-duty days with their employment as a firefighter, and one off-duty day) for standardization among the various shift schedules. Firefighters on the 4 s and 6 s were asked to complete their food log during a rotation of on-off-on duty, and the firefighters on the 48–96 to record on-on-off duty. This standardization allowed two on-duty days and one-off duty day to be recorded for all participating firefighters. Upon completing the three-day food record, reports were emailed to the principal investigator. The report was imported into the ESHA Food Processor Software to analyze average nutrient consumption.

Data analyses were performed using the Statistical Package for Social Sciences (Armonk, NY USA, version 24). Descriptive statistics (means, standard deviations, 95% confidence intervals) were calculated for all participant characteristics and the average three days nutrient intakes. For each nutrient variable, the primary analysis was a comparison of 95% confidence intervals (CIs) for each variable of recorded dietary intake to the MDRI reference value. Dietary intake that fell within or outside of the MDRI reference value at 95% CI were noted accordingly, with significance accepted at *p* = 0.05. Additionally, participants were categorized into four groups for fiber, fatty acids, and micronutrients: 0–33%, 34–67%, 68–99%, and ≥100% of the MDRI, and were categorized into three groups for the macronutrients (see Figure 1 and Figure 2).

## 3. Results

All participants (*n* = 150) were males and key baseline characteristics reflect some diversity, and generally reflect the population of firefighters from which the sample was derived (Table 1). The age of participants was 37.4 ± 8.4 years (min.: 22; max.: 59 years), BMI was 28.3 ± 3.7 kg/m^2^ (min.: 21.3 kg/m^2^; max: 42.9 kg/m^2^), and % bodyfat was 21.5 ± 6.0% (min.: 8.6%; max: 41.2%).

Comparison of dietary intake using the 95% confidence interval to MDRI reference value for each variable is found in Table 2. Firefighters in this study consumed below the MDRI standard for total calories, percentage of total calories from carbohydrates, linoleic and alpha-linolenic fatty acid, fiber, vitamin D and E, folate, calcium, magnesium, potassium, and zinc. They consumed above the MDRI standard for percentage of total calories from fat, vitamin A, riboflavin, niacin, vitamins B6 and B12, iron, phosphorus, selenium, and sodium. When compared between nutrients, frequency statistics reveal that vitamin D, magnesium, and potassium levels had the greatest shortcomings to the MDRI standard (95.3%, 94.0%, and 98.7%, respectively, under 100% of MDRI). These nutrients were largely under-consumed by the firefighters, and most did not meet the MDRI. From this group, only 4.7% of the firefighters met or exceeded vitamin D MDRI. Magnesium intake meeting 100% or greater of the MDRI was only met by 6% of the firefighters. The reference intake for potassium was only met by 1.3% of the population for both the DRI and MDRI. Approximately 80% of the sample consumed greater than the MRDA for sodium, and only 5.3% (8/150) of firefighters consumed at least the MDRI for calories (3400 daily calories). The majority of firefighters (129/150; 86%) were above the MDRI range for the percentage of calories from fat, and were below the MDRI range for the percentage of calories from carbohydrates (135/150; 90%). Most firefighters (144/150; 96%) were within the MDRI range for percentage of calories from protein. When compared to the MRDA for grams of protein, carbohydrates, and fat, firefighters consumed adequate amounts of protein, inadequate amounts of carbohydrates, and were within the maximum allowance of fat.

## 4. Discussion

The purpose of this study was to assess nutrient intakes of career firefighters and compare averages to the MDRI standards. The study was successful in utilizing MDRI as a comparison marker for a sample of career firefighters (*n* = 150) in Southern California. To our knowledge, firefighters’ nutritional intake has not been compared to MDRI. The primary finding of this study is that the majority of career firefighters are not meeting the MDRI standards in several key nutrient variables that are required to promote health, improve performance, enhance recovery and resiliency, and reduce chronic disease for tactical demands. Similar results were found from our sample population of firefighters and military population when comparing nutrient intake to MDRI. Lutz et al. [10] found male military recruits under-consumed alpha-linolenic and linoleic fatty acids, fiber, vitamin D and E, magnesium, and potassium. Previous research with this sample found 95% of firefighters were at moderate or high nutritional risk based on the Tactical Athlete Nutrition Score [19], and the majority were under-consuming calories based on the recommended caloric intake [19], which is generally consistent with the current study.

The findings from the current study suggest that firefighters are consuming diets adequate in vitamin A, most B-vitamins, vitamin C, iron, phosphorus, selenium, and meeting the standards for protein as a percentage of total calories. Similar results were observed with previous research in firefighters regarding nutrient intake using FFQ. Pohle-Krauza et al. [7] found high levels of sodium and low levels of fiber consumption compared to the Dietary Guidelines for Americans [20]. In the current study, firefighters had high levels of sodium intake and low levels of fiber, according to the MDRI standards [5]. Protein intake from self-reported three-day food records in overweight and obese career firefighters averaged 17% of total calories [14]. In our sample, the mean BMI was 28.3 kg/m^2^ and protein intake averaged 21.9% of total calories. Consistent with the findings of the current study, Robertson et al. [8] found firefighter populations overconsuming sodium, and under-consuming fiber, potassium, and vitamins D and E. The effects of inadequate micronutrients in firefighter populations have not been studied. It is important to note the food culture of the fire service typically involves being family-style with large portions; thus, providing education is a pathway to improving nutrient intake with the population [21]. The nutrients being inadequately consumed have documented benefits related to health and performance that might be beneficial for firefighting. Vitamin E acts as an antioxidant to reduce free radicals from daily physical activities, while vitamin D acts as a hormone and plays a role in muscle synthesis [22]. Magnesium and potassium are associated with muscle cramps, and magnesium and calcium play a role in bone health. All three minerals are part of the Dietary Approach to Stop Hypertension, which has been shown to be cardio-protective [13]. Firefighter populations have a high prevalence of injuries, obesity, and cardiovascular disease.

Professional firefighting is classified as a “heavy” job demand, according to the U.S. Department of Labor Physical Demand Characteristics of Work levels [23]. The typical amount of energy required for a job characterized as “heavy” is 6.4–7.5 metabolic equivalents (METS) [24]. The sample of professional firefighters enrolled in this study were expected to perform at these levels as a condition of their employment at the departments. In addition to the base level of job demands and related energy expenditure, the participants’ self-reported physical activity levels were assessed in this study according to the recommendations of the American College of Sports Medicine [25]. We found that that 72.0%, 83.3%, and 64.7% of the participants were meeting or exceeding the recommended quantity of exercise for cardiovascular exercise, resistance exercise, and flexibility exercise, respectively. Thus, the sample of firefighters enrolled in this study have high energy requirements for their jobs and are physically active outside of the job demands. Thus, we believe the energy requirements for this sample of firefighters exceeds that of the general population and is comparable to military personnel.

This study represents a continuation of a series of research comparing two tactical athletes’ populations—firefighters and military personnel. For example, research in other areas of health and wellness found that firefighters and active duty military personnel were similar in size (cross-sectional area) with regard to the low back and abdominal musculature [26]. Another study demonstrated that active duty military personnel had significantly higher back and musculature endurance [27]. MDRI has not been formally compared between firefighters and the military. However, previous work in the military has shown that military personnel are meeting vitamin B6 and B12, as well as protein requirements, but are not meeting their requirements for alpha-linolenic and linoleic fatty acid, fiber, vitamin D and E, magnesium, and potassium for the MDRI standards [10] like firefighters.

The potential limitations of the current study include the following. (1) This study compared firefighters to military-specific nutrient needs (MDRI). It is unclear if MDRI is the best marker to assess firefighter nutritional intake. However, there are no other DRI available for firefighters. Given the lack of scientific evidence and consensus guideline statements on this topic, the findings of this study provide a possible framework for assessing nutrient intake in firefighters. (2) Job performance demands are different between the military and firefighting. However, the occupational demands of professional firefighting are much more closely aligned with the military than the general population. (3) The convenience sample selected may not represent all career firefighters in the U.S. The sample population was from Southern California, and dietary intakes may differ compared to their colleagues across the country. However, the sample of firefighters enrolled in this study were from a diverse group of 13 fire departments, ranging from urban, coastal, native America reservation, and the national border. Thus, the findings of this study can be generalized to professional firefighters for departments with similar characteristics as those assessed in this study. (4) This study did not ask firefighters to report vitamin/mineral supplements; however, micronutrients consumed from whole foods have synergistic effects, such as phytochemicals and fiber, not found in multivitamins. To our knowledge, no peer-reviewed data are available describing the specific vitamin/mineral supplement intake of firefighters. Future research should consider addressing supplementation in firefighters. (5) This study also assumed that the firefighters accurately self-reported their dietary intakes and portions following the protocol. Nevertheless, a strength of this study is the use of the three-day food record procedure for which reliability has been established [9] to allow specific nutrient comparison. Further, a Registered Dietitian inquired about nutrient intake at subsequent consultations with participants who were deemed outliers regarding nutrient intake. We believe that this two-pronged approach allowed for a reliable and valid method to assess nutrient intake for the 150 participants.

Potential sources of bias in data collection and the selected outcome measurement include the following. Participants were asked to continue their normal eating patterns while recording their food and beverage intake to reduce bias. However, self-reported dietary intakes can be biased because individuals often under-report total energy intakes [28] and a one-time snapshot. During analysis, the investigator screened the food records to ensure accuracy with the days logged. One firefighter was excluded because the food record only included one day. Approximately 19% (150/800) of the target population and 44% (150/341) of those enrolled in the Regional Wellness Initiative participated in this study. While these numbers may indicate selection bias, the analyses suggest that there are no inherent differences in the study sample and the population from which it was derived. To assess whether there were selection biases or demographic differences (e.g., obese firefighters not recording food) for those who completed a food record (*n* = 150) and those who did not (*n* = 190), an independent *t-*test was run to compare groups among key demographic, health, and fitness variables. The *t*-test analyses for all variables compared both groups. No significant differences were observed between those who completed the three-day food record and those who did not for age, BMI, WC, BF%, physical fitness measures (muscular endurance and Functional Movement Screen), history of low back pain, or current low back pain. This suggests no selection bias in the sample of firefighters who completed three day food records. In other words, one group (those who completed food record versus those who did not) was not more obese, younger, or more physically fit than the other.

The findings of this study echo the need to develop dietary reference intakes specific to firefighting tasks. Future research is needed to determine if specific nutrient inadequacies negatively impact health and performance in career firefighters. Furthermore, large-scale studies are necessary to assess which specific nutrients are needed to develop a Firefighter Dietary Reference Intake, or whether MDRI is adequate for assessing firefighters. Randomized controlled trials are needed to assess the efficacy of various interventions on improving nutrient intake in firefighters, particularly in deficit areas identified by DRIs. Dietitians play a critical role in counseling this population to improve nutrient intakes and may use the results of this study to assess dietary intake in career firefighters. In the absence of validated methods to assess firefighter dietary intake, the three-day food record for comparison to MDRI is a reasonable approach to help guide the implementation of customized nutritional programs in firefighters.

## 5. Conclusions

This study found that the majority of career firefighters are not meeting the MDRI standards in several key nutrient variables that are required to promote health, improve performance, enhance recovery, and reduce chronic disease for tactical demands. Dietitians may use the results of this study to help design customized nutrition programs for this population. Given the unique job demands of firefighters compared to those in the military, future research should develop customized reference intakes for firefighters.

## Figures and Tables

**Figure 1 nutrients-12-01876-f001:**
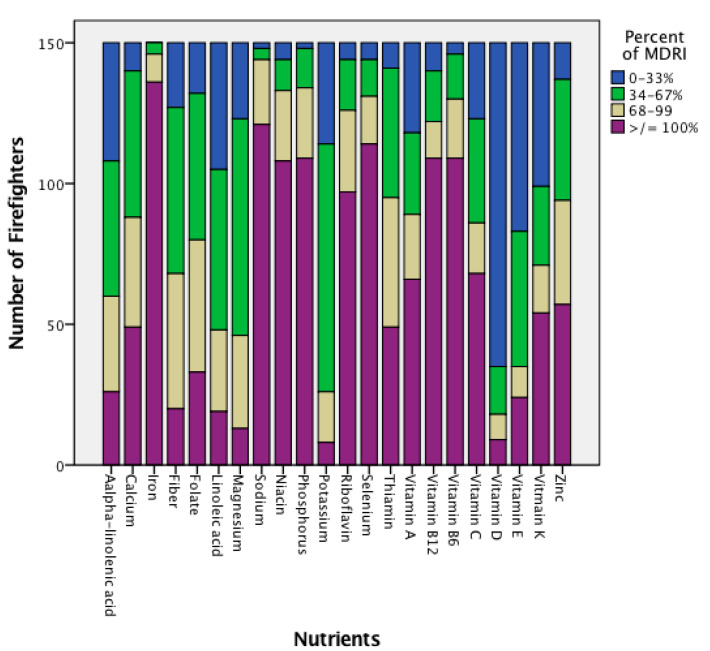
Categorized three-day average nutrient intake of firefighters compared to Military Dietary Reference Intake [5].

**Figure 2 nutrients-12-01876-f002:**
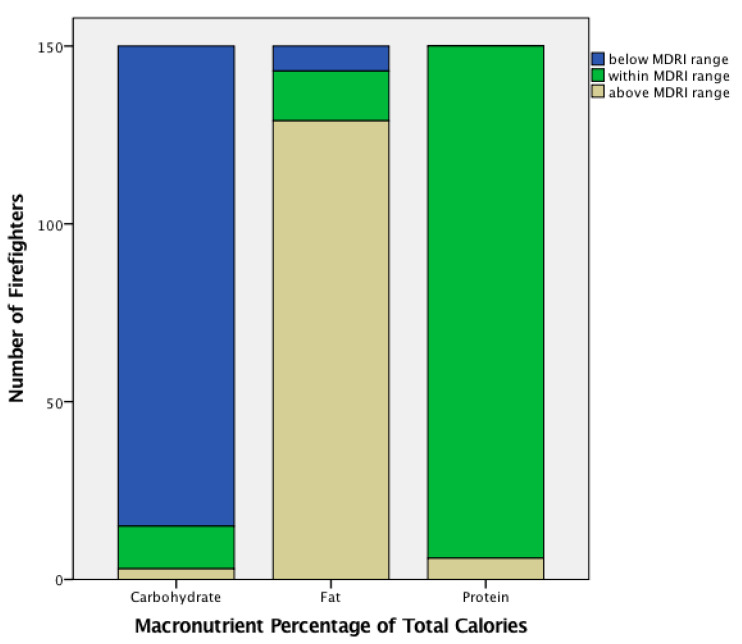
Categorized three-day average for macronutrient intake of firefighters compared to MDRI [2].

**Table 1 nutrients-12-01876-t001:** Demographic characteristics of participants.

Variable	*n* Valid	*n* Missing	Mean	SD	Min	Max
Age (years)	150	0	37.35	8.44	22	59
Resting Heart Rate (BPM)	149	1	64.66	11.1	46	96
Blood Pressure Systolic (mmHg)	149	1	129.82	14.29	103	191
Blood Pressure Diastolic (mmHg)	149	1	76.87	9.92	52	107
Body Mass Index (kg/m^2^)	150	0	28.25	3.65	21.3	42.9
Waist Circumference (in)	149	1	93.61	9.65	76.5	118.7
Body Fat (%)	148	2	21.54	5.98	8.6	41.2

Key: SD = Standard Deviation; Min = Minimum; Max = Maximum; BPM = Beats Per Minute.

**Table 2 nutrients-12-01876-t002:** Average three-day nutrient intake compared to DRI and MDRI standards.

Variable	DRI	MDRI (Male)	Daily Intake		
			Mean	SD	95% CI (Min, Max)
Calories (kcal/day)		3400	2292	630.4	2182.3, 2401.9 *
Protein (%)	10–35	10–35	21.9	6.2	20.8, 22.8
Protein (g/day)		102	123.4	45.3	116.2, 130.6
Carbohydrate (%)	45–65	50–55	37	10	35.4, 38.6 *
Carbohydrates (g/day)		510	209.5	79.2	196.8, 221.8 *
Fat (%)	20–35	25–30	39.4	8.6	38.0, 40.8 *
Fat (g/day)		<113	101.5	39.6	95.2, 107.8
Linoleic acid (g/day)	17	17	9.6	6.3	8.6, 10.6 *
Alpha-linolenic acid (g/day)	1.6	1.6	1.1	0.9	0.9, 1.2 *
Fiber (g/day)	38	34	21.9	10.2	20.0, 23.5 *
Vitamin A (μg/day)	900	900	914.1	786.9	787.1, 1041.0
Vitamin D (IU/day)	600	600	220.1	668.9	112.1, 327.9 *
Vitamin E (mg/day)	15	15	7.5	7.4	6.4, 8.7 *
Vitamin K (μg/day)	120	120	113.8	121.2	94.2, 133.4
Thiamin (mg/day)	1.2	1.2	1.1	0.7	0.9, 1.2
Riboflavin (mg/day)	1.3	1.3	1.7	1	1.5, 1.9 *
Niacin (mg/day)	16	16	23.2	12.5	21.2, 25.2 *
Vitamin B6 (mg/day)	1.3	1.3	1.9	1.3	1.7, 2.2 *
Vitamin B12 (μg/day)	2.4	2.4	4.7	4	4.0, 5.3 *
Folate (μg/day)	320	400	300.4	222.1	264.6, 336.3 *
Vitamin C (mg/day)	75	90	114.7	186.2	84.7, 144.8
Calcium (mg/day)	800	1000	852.1	434.9	781.9, 922.2 *
Iron (mg/day)	6	8	15.4	9.5	13.8, 16.9 *
Magnesium (mg/day)	400	420	232.7	117.2	213.8, 251.6 *
Phosphorus (mg/day)	580	700	1018.7	483.8	940.7, 1096.8 *
Potassium (mg/day)	4700	4700	2201.3	933.5	2050.7, 2351.9 *
Selenium (μg/day)	45	55	93.5	53.1	84.9, 102.1 *
Sodium (mg/day)	1500	<2300	3442.9	1361.5	3223.3, 3662.6 *
Zinc (mg/day)	9.4	11	9.7	5.5	8.8, 10.5 *

Key: * *p* < 0.05: Daily Intake compared to MDRI; Protein (%), Carbohydrate (%), Fat (%) = % of total calories for each macronutrient; Daily intake = Average three-day nutrient intake; DRI = Dietary Reference Intake; MDRI = Military Dietary Reference Intake; SD = Standard Deviation; 95% CI = 95% Confidence Interval.

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
