# Peer review of "Evaluating Nutrient Intake of Career Firefighters Compared to Military Dietary Reference Intakes"

_nutrients, 2020, doi:10.3390/nu12061876_

Round 1

Reviewer 1 Report

Great job. I do agree that much research has focused on intervention-based studies with FF and little is known about the adequacy of nutrient intakes compared to standard DRI or MDRI. Firefighters are considered industrial athletes and evaluating their nutrient intake based on their high job demands should be a priority in order to ensure that they will be performing their duties safer, while they are fit for duty from all different persepctives.

Author Response

Thank you for recognizing the importance of comparing firefighters’ nutrient intake to the MDRI. While you did not have any suggested changes, we have made changes based on the other reviewers’ comments. Please see the revised manuscript for approval.

Reviewer 2 Report

This investigation aimed to compare the dietary intake of a sample of firefighters to the MRDI. While the paper is generally well-written, there are major concerns. You mention several limitations in the Discussion, and I am afraid that the strength of these limitations is too severe.

Major:

-As you state, the military provides the only published guidelines for 'tactical athletes', however, the MRDI does not explicitly state that these guidelines are meant for tactical athletes, but rather for military personnel only. Although there are several similarities in job-related demands of firefighters and military personnel, they are not the same. Therefore, dietary intakes of firefighters probably should not be compared to a standard reference designed specifically for military personnel.

-The MRDI was updated in 2017, but it seems like you have used the old MRDI in this analysis.

-The new MRDIs were designed specifically for men and women between 19-30 years old. The average age of participants in your study was ~37 years. As requirements change based on age, the MRDI may not be appropriate for this population.

-Physical activity levels will affect dietary intake requirements. You have not reported physical activity habits, but this is imperative to make accurate comparisons.

Minor:

-Please describe how the demographic characteristics of participants were obtained in the Materials and Methods.

-All references need to be examined carefully and adjusted to fit the journal style.

-Why were only 3 groups used for analysis of macronutrients but 4 groups were used in the analysis of micronutrients? You may want to reconsider keeping it homogenous between analyses.

-The first paragraph of the Results section fits better in the Materials and Methods section.

Author Response

Comment-As you state, the military provides the only published guidelines for 'tactical athletes', however, the MRDI does not explicitly state that these guidelines are meant for tactical athletes, but rather for military personnel only. Although there are several similarities in job-related demands of firefighters and military personnel, they are not the same. Therefore, dietary intakes of firefighters probably should not be compared to a standard reference designed specifically for military personnel.

  • Reply- We added a deeper discussion on why the limitations and comparison of firefighters’ nutrient intake is appropriate. For example, we added “The potential limitations of the current study include: 1. This study compared firefighters to military-specific nutrient needs (MDRI). It is unclear if MDRI is the best marker to assess firefighter nutritional intake. However, there are no DRI available for firefighters. Given the lack of scientific evidence and consensus guideline statements on this topic, the findings of this study provide a possible framework for assessing nutrient intake in firefighters.2. Job performance demands are different between military and firefighting. However, the occupational demands of professional firefighting are much more closely aligned with military than general population.” Further, we added several sentences to the limitation discussion, see lines 239-259.

Comment-The MRDI was updated in 2017, but it seems like you have used the old MRDI in this analysis.

  • Reply: Thank you for the note on the updated 2017, we did have that update but the citation software did not update from the older version. The current analysis was compared to the 2017 MDRI. We updated all the citations within the manuscript to reflect the updated citation.

Comment-The new MRDIs were designed specifically for men and women between 19-30 years old. The average age of participants in your study was ~37 years. As requirements change based on age, the MRDI may not be appropriate for this population.

  • Reply- The age difference is appropriate to question. However, the DRI for 19-30 and 31-50 are the same for micronutrients, macronutrient percentage of calories, and total calories. The MDRI energy requirements for 19-30 are based on average military personnel at 187 pounds and 69 inches, our sample has an average weight of 201 pounds and 70 inches, which would require more energy to maintain weight. Therefore, the age does not change the daily nutrient requirements, making MDRI appropriate for the sample population based on age.

Comment-Physical activity levels will affect dietary intake requirements. You have not reported physical activity habits, but this is imperative to make accurate comparisons.

  • Reply- We added a paragraph regarding physical activity and the data we collected. See lines 215-227.

“Professional firefighting is classified as “heavy” job demand according to the US Department of Labor Physical Demand Characteristics of Work levels.24 The typical energy required for a job characterized as “heavy” is 6.4 - 7.5 metabolic equivalents (METS).25 The sample of professional firefighters enrolled in this study were expected to perform at these levels as a condition of their employment at the departments. In addition to the base level of job demands and related energy expenditure, the participants’ self-reported physical activity levels were assessed in this study according to the recommendations of the American College of Sports Medicine.26 We found that that 72.0%, 83.3%, and 64.7% of the participants were meeting or exceeding the recommended quantity of exercise for cardiovascular exercise, resistance exercise, and flexibility exercise, respectively. Thus, the sample of firefighters enrolled in this study have high energy requirements for their jobs and are physically active outside of the job demands. Thus, we believe the energy requirements for this sample of firefighters exceeds that of the general population and is comparable to military personnel.”

Minor:

Comment-Please describe how the demographic characteristics of participants were obtained in the Materials and Methods.

  • Reply: We added a sentence about demographic characteristics under Materials and Methods, See lines 105-110. “After consent, participants completed a battery of baseline assessments, which included self-report questionnaires, anthropometric measures, and physical fitness tests as part of the overlying grant. Self-report questionnaires for demographics, health history, injury and illness, physical function, disability, behavioral, physical activity, exercise habits, diet habits, stages of change, sleep questionnaires, and a food record were obtained.”

Comment-All references need to be examined carefully and adjusted to fit the journal style.

  • Reply- references were updated to ACS format per the manuscript required format.

Comment-Why were only 3 groups used for analysis of macronutrients but 4 groups were used in the analysis of micronutrients? You may want to reconsider keeping it homogenous between analyses.

  • Reply- 3 groups were analyzed for macronutrients because the macronutrients have an established range (e.g. 10-35% protein), whereas, the micronutrients have a set point (e.g. 800 mg calcium), therefore, we chose to only have 3 groups for the macronutrients: 1. Below the range, 2. Within the range, and 3) above the range.

Comment-The first paragraph of the Results section fits better in the Materials and Methods section.

  • Reply- We moved the paragraph under Materials and Methods. See lines 138-147.

Reviewer 3 Report

Line 11: change general public to public

Keywords: change first-responders to first responders

Line 31: Consider changing “goal” to “goals” as you list multiple primary outcomes or the DRI

Line 37: I would suggest that most professional require physical training for occupational performance. I believe what the authors are trying to suggest is that they have to be physically fit to do their job, but the unique factor is they are public servants who must wear loads of external gear and supplies to do their job. The physical training is one aspect and believe this sentence should be revised.

Line 43: consider removing “likely” before not

Line 45: Using the term “gap” here implies you had a preconceived notion that the data would support a gap. I would suggest altering this wording.

Line 48: I like the authors connection to food records (or food logs) but not all are 3-days. I would suggest exploring why 3-day is “the best” or suggested here in the paper.

Line 75: Were these structural firefighters? Given the study was based in Southern California, the likelihood you had wildland firefighters is high. If this was the case, the work output demands between structural and wildland firefighters is vastly different due to the environment and uniform unique to each job sector. Please revise to include this information. If they were structural, were these firefighters on 24 hr or 48 hr shifts? This is important for the connection to the methods.

Line 94-98: The reason I was asking about shifts for the participants is connected back to the 24 hr recall and 3 day measure of dietary intakes. Was this collected while the participant was on duty including the 24 hr before or was this period their “off time” which influenced their ability to eat as they wish as they did not have to work/perform in that time frame? I believe you come back to this in Line 112 but I think this all needs to be a little more clear for a reader about the WHY of this who are unfamiliar with the firefighter routine.

Line 117-122: This paragraph is good info but seems very oddly placed. Consider moving to combine with first paragraph of this section

Line 124: SPSS has changed locations – now based out of Armonk, NY and owned by IBM. Please revise.

Line 128: Would suggest revising this sentence that starts with 95% to not start it with a number

Line 131: Seems to be a grey background on the > sign before 100%

Line 159: Revise sentence “most firefighters of were within the”

Line 188-190: These findings about sodium intake are interesting. I would suggest connecting this back to the lifestyle based off the shift work of a firefighter where they often cook and eat “family” style in the fire house. How does this affect and alter the dietary intake? This is vastly different than the military that has a specific diet prepared by the base especially during basic training. How can you make some connections back to this? To me, the firefighters must understand this information about easily made, family style meals to ensure their dietary habits change.

Table 2 – Can you sort the table by meeting standards and not meeting? Or highest to lowest? Some organization would be very helpful here.

Line 198: change “athlete” to “athletes”

Line 211: Remove “likely”

Line 223: Change “snap-shot” to “snapshot”

Some references are incorrectly presented - please review these.

Author Response

Line 11: change general public to public

  • Reply: We changed general public to public.

Keywords: change first-responders to first responders

  • Reply: We removed the – for the key words.

Line 31: Consider changing “goal” to “goals” as you list multiple primary outcomes or the DRI

  • Reply: We added an s to goals for this sentences.

Line 37: I would suggest that most professional require physical training for occupational performance. I believe what the authors are trying to suggest is that they have to be physically fit to do their job, but the unique factor is they are public servants who must wear loads of external gear and supplies to do their job. The physical training is one aspect and believe this sentence should be revised.

  • Reply: We added an additional sentence to address your comment, see lines 37-39. “Firefighting is one of the most hazardous, physically demanding, and psychologically stressful occupations, requiring physical exertion while wearing heavy protective gear.”. We also included information on current physical habits and demands on firefighting, see lines 215-227.

Line 43: consider removing “likely” before not

  • Reply: We removed the word likely in this sentence.

Line 45: Using the term “gap” here implies you had a preconceived notion that the data would support a gap. I would suggest altering this wording.

  • Reply: We changed the word “gap” to habits. The new sentence now reads “Thus, research is needed to identify nutritional habits in firefighters’ diets so that nutrition interventions can be developed to improve firefighters’ quality of life and job performance.” See lines 46-48.

Line 48: I like the authors connection to food records (or food logs) but not all are 3-days. I would suggest exploring why 3-day is “the best” or suggested here in the paper.

  • Reply: We added justification for 3-days instead of longer durations stating “Three-day food records are deemed valid of average intakes when compared to longer durations and reduces survey fatigue.” See lines 51-52.

Line 75: Were these structural firefighters? Given the study was based in Southern California, the likelihood you had wildland firefighters is high. If this was the case, the work output demands between structural and wildland firefighters is vastly different due to the environment and uniform unique to each job sector. Please revise to include this information. If they were structural, were these firefighters on 24 hr or 48 hr shifts? This is important for the connection to the methods.

  • Reply: We added more information about the department shift and schedule for clarification- The study was conducted with 13 career fire departments in Southern California, US ranging from urban, coastal, native American reservation, and national border departments. Full duty, career firefighters (n = 150, males) were enrolled in this study. The shift schedule within the participating fire departments were either 4s and 6s (24-hour shift) or 48-96 (48-hour shift). See lines 83-87.

Line 94-98: The reason I was asking about shifts for the participants is connected back to the 24 hr recall and 3 day measure of dietary intakes. Was this collected while the participant was on duty including the 24 hr before or was this period their “off time” which influenced their ability to eat as they wish as they did not have to work/perform in that time frame? I believe you come back to this in Line 112 but I think this all needs to be a little more clear for a reader about the WHY of this who are unfamiliar with the firefighter routine.

  • Reply: We added more information about the department shift and schedule for clarification- Firefighters on the 4s and 6s were asked to complete their food log during a rotation of on-off-on duty, and the firefighters on the 48-96 to record on-on-off duty. This standardization allowed 2 on-duty days and 1-off duty day to be recorded for all participating firefighters. See lines 132-135.

Line 117-122: This paragraph is good info but seems very oddly placed. Consider moving to combine with first paragraph of this section

  • Reply: We moved the paragraph to the beginning of the section, see lines 76-81.

Line 124: SPSS has changed locations – now based out of Armonk, NY and owned by IBM. Please revise.

  • Reply: We updated the location within the citation for SPSS.

Line 128: Would suggest revising this sentence that starts with 95% to not start it with a number

  • Reply: We reordered the sentence to say “Dietary intake that fell within or outside of the MDRI reference value at 95% CI were noted accordingly with significance accepted at p = 0.05.” See lines 142-143.

Line 131: Seems to be a grey background on the > sign before 100%

  • Reply: We removed the grey background.

Line 159: Revise sentence “most firefighters of were within the”

  • Reply: This sentence was revised to say “most firefighters were within the”

Line 188-190: These findings about sodium intake are interesting. I would suggest connecting this back to the lifestyle based off the shift work of a firefighter where they often cook and eat “family” style in the fire house. How does this affect and alter the dietary intake? This is vastly different than the military that has a specific diet prepared by the base especially during basic training. How can you make some connections back to this? To me, the firefighters must understand this information about easily made, family style meals to ensure their dietary habits change.

  • Reply: We added a sentence about the culture of firehouse meals under discussion stating “It is important to note the food culture of the fire service typically involves family style with large portions; providing education is a pathway to improving nutrient intake with the population.” See lines 206-208.

Table 2 – Can you sort the table by meeting standards and not meeting? Or highest to lowest? Some organization would be very helpful here.

  • Reply: We did not reorganize that graph because SPSS randomly sorts it.

Line 198: change “athlete” to “athletes”

  • Reply: We added an s to athlete for plural grammar.

Line 211: Remove “likely”

  • Reply: We removed the word likely from this sentence.

Line 223: Change “snap-shot” to “snapshot”

  • Reply: The word was changed.

Some references are incorrectly presented - please review these.

  • Reply: We have updated the references to match the Nutrients required format of ACS.

Round 2

Reviewer 2 Report

Thank you for addressing my comments and suggestions. I believe that the paper has been improved.

-One of my main comments in the last round of reviews was that you were not using the correct MRDI 2017 reference standards. For example, the overall energy (calorie) intake for the average male is listed at 3400 under 'moderate' activity in the 2017 MRDI, however you report 3200 in Table 2. Additionally, you report incorrect reference values for zinc, sodium, magnesium, riboflavin, vitamin C, and several other variables listed. Therefore, your current results do not accurately reflect a comparison to the current MRDI. Please redo your analysis entirely to reflect the correct MRDI values. This will likely cause significant changes to your results, figures, and tables.

-"MRDA" in Table 2 should be "MRDI".

-Please review the manuscript extensively for punctuation edits.

Author Response

Thank you for the quick review. We had previously submitted to another journal who pointed out we did not have the most up-to-date MDRI values. We re-analyzed the values and wrote the paper based on those updates. That manuscript required the table to be separate from the main document. I copied the wrong table into the current manuscript we submitted to Nutrients. Since your second review, the table was correct with the values that were to analyze and create the figures. I went back to our SPSS output to ensure the nutrients were compared to the current 2017 MDRI levels and put into the categorical variables (for the figure) and frequencies. They were all correct and based on the MDRI 2017 references. The mistake I made for both submission was not putting the updated 2017 table into this manuscript. That has been updated (see attached). None of the analysis changed, except the frequency of calories consumed, which has been noted in line 154. 

The punctuations have been corrected, specifically all of the periods were placed after the reference. 

Round 3

Reviewer 2 Report

I have no further comments.